# Infiltration of CD3+ and CD8+ lymphocytes in association with inflammation and survival in pancreatic cancer

**Gerik W. Tushoski-Alemán**[1], **Kelly M. Herremans**[1], **Patrick W. Underwood**[1], **Ashwin Akki**[2], **Andrea N. Riner**[1], **Jose G. Trevino**[1], **Song Han**[1], **Steven J. Hughes**[1]*

1 Department of Surgery, College of Medicine, University of Florida, Gainesville, Florida, United States of America, 2 Department of Pathology, College of Medicine, University of Florida, Gainesville, United States of America

* Steven.Hughes@surgery.ufl.edu

**Data Availability Statement:** All relevant data are within the paper and its Supporting information files.

## Abstract

### Background

Pancreatic ductal adenocarcinomas (PDAC) have heterogeneous tumor microenvironments relatively devoid of infiltrating immune cells. We aimed to quantitatively assess infiltrating CD3+ and CD8+ lymphocytes in a treatment-naïve patient cohort and assess associations with overall survival and microenvironment inflammatory proteins.

### Methods

Tissue microarrays were immunohistochemically stained for CD3+ and CD8+ lymphocytes and quantitatively assessed using QuPath. Levels of inflammation-associated proteins were quantified by multiplexed, enzyme-linked immunosorbent assay panels on matching tumor and tissue samples.

### Results

Our findings revealed a significant increase in both CD3+ and CD8+ lymphocytes populations in PDAC compared with non-PDAC tissue, except when comparing CD8+ percentages in PDAC versus intraductal papillary mucinous neoplasms (IPMN) (p = 0.5012). Patients with quantitatively assessed CD3+ low tumors (lower 50%) had shorter survival (median 273 days) compared to CD3+ high tumors (upper 50%) with a median overall survival of 642.5 days (p = 0.2184). Patients with quantitatively assessed CD8+ low tumors had significantly shorter survival (median 240 days) compared to CD8+ high tumors with a median overall survival of 1059 days (p = 0.0003). Of 41 proteins assessed in the inflammation assay, higher levels of IL-1B and IL-2 were significantly associated with decreased CD3+ infiltration (r = -0.3704, p = 0.0187, and r = -0.4275, p = 0.0074, respectively). Higher levels of IL-1B were also significantly associated with decreased CD8+ infiltration (r = -0.4299, p = 0.0045), but not IL-2 (r = -0.0078, p = 0.9616). Principal component analysis of the inflammatory analytes showed diverse inflammatory responses in PDAC.

**Funding:** This research was funded by National Cancer Institute (https://www.cancer.gov/) (R01CA242003 to JGT, U54CA233444 to JGT, and U54CA233444-03S1 to ANR and JGT) and National Institute of Diabetes and Digestive and Kidney Diseases of the National Institutes of Health (https://www.niddk.nih.gov/) under award number 1U01DK108320 (SJH) and the Joseph and Ann Matella Fund for Pancreatic Cancer Research (https://www.uff.ufl.edu/giving-opportunities/023909-joseph-and-ann-matella-fund-for-pancreatic-cancer-research/) (JGT). Authors also receive salary support from the National Human Genome Research Institute of the National Institute of Health (https://www.genome.gov/) (T32 HG008958 to K.M.H., and A.N.R.). Authors are also supported by the Collaborative Alliance of Pancreas Education and Research (https://www.caper-pancreas.org/) (K.M.H. and A.N.R.). The funders had no role in study design, data collection and analysis, decision to publish, or preparation of the manuscript.

**Competing interests:** The authors have declared that no competing interests exist.

**Abbreviations:** IPMN, Intraductal papillary mucinous neoplasm; PCA, Principal component analysis; PDAC, Pancreatic ductal adenocarcinoma; PNET, Pancreatic neuroendocrine tumor; TMA, Tissue microarray; TME, Tumor microenvironment.

## Conclusion

In this work, we found a marked heterogeneity in infiltrating CD3+ and CD8+ lymphocytes and individual inflammatory responses in PDAC. Future mechanistic studies should explore personalized therapeutic strategies to target the immune and inflammatory components of the tumor microenvironment.

## Background

Pancreatic ductal adenocarcinoma (PDAC) exhibits complex molecular and cellular heterogeneity, and clinical outcomes remain poor [1]. Compared to other tumor types, PDAC is considered an immune-cold cancer, escaping immune recognition by cytotoxic T-cells (CD8+), and other anti-tumor immune cell types. It is also resistant to immune checkpoint inhibition monotherapy [2,3]. Complex inflammatory processes have been shown to modulate PDAC progression [4,5]. For example, we and others have demonstrated that cancer associated fibroblasts (CAFs) impact the tumor microenvironment and foster tolerance [6,7]. Although PDAC is typically described as a non-immunogenic tumor, reprogramming the dense desmoplastic reaction, and resulting inflammatory response may offer therapeutic opportunities for patients.

The activation and recruitment of CD3+ and CD8+ lymphocytes are fundamental to mounting an effective tumor response in nearly all cancers. Various techniques have been employed to evaluate lymphocyte populations. Semi-quantitative approaches are routinely applied in clinical and diagnostic settings. Quantitative and automated approaches have been shown to enable standardized analysis of large datasets, enhanced reproducibility, and precision in anticipating immunotherapy response [8–10]. These approaches have not yet been fully assessed or compared in PDAC. Neoadjuvant therapy has been shown to alter the immune microenvironment of pancreatic cancers [11,12], and as such, a treatment naïve cohort may offer enhanced insight to true lymphocyte heterogeneity in PDAC. We hypothesize that a quantitative approach could offer improved accuracy in assessing lymphocyte infiltration for overall survival in pancreatic cancer patients. We further hypothesize that infiltrating T-cells are associated with inflammatory proteins in the tumor microenvironment. We anticipate that associations between infiltrating T-cells and inflammatory proteins in the tumor microenvironment may shed light on the underlying mechanisms of T-cell recruitment and activation to provide potential targets to improve therapeutic efficacy in PDAC.

In this study, we assembled tissue-microarrays (TMAs) of treatment naïve PDAC tumor and non-PDAC controls. We quantitatively determined intratumoral CD3+ and CD8+ lymphocyte populations. The design of this study augments previous studies that have relied on semi-quantitative assessment and included neoadjuvant treated PDAC patients [13]. We performed multiplex analysis on matching tumors to assess relationships between immune cell infiltrations and inflammatory proteins in the tumor microenvironment. We also identified multiple inflammatory factors as indicators of high CD3+ or CD8+ involvement.

## Methods

### Patient cohort and tissue preparation

All studies were approved by the Institutional Review Board at the University of Florida (IRB201600873) and informed consent was obtained from all patients prior to any tissue collection. Patients were recruited into the study from May 2012 to January 2022. This study was conducted from January 2017 to March 2023. Authors had access to information that could

**Table 1. Clinicopathological characteristics of patient cohort.**

| | | PDAC (n = 59) | Benign (n = 37) |
|---|---|---|---|
| **Age, mean (SD)** | | 69.93 (9.3) | 52.59 (14.8) |
| **Race, n (%)** | **White** | 57 (96.6) | 32 (86.5) |
| | **African American** | 1 (1.7) | 5 (13.5) |
| | **Other** | 1 (1.7) | 0 (0) |
| **Sex, n (%)** | **Male** | 36 (61.0) | 13 (35.0) |
| | **Female** | 23 (39.0) | 24 (65.0) |
| **Pathology, n (%)** | **PDAC** | 59 (100.0) | NA |
| | **Chronic pancreatitis** | NA | 20 (54.1) |
| | **Intraductal papillary mucinous neoplasm (IPMN)** | NA | 6 (16.2) |
| | **Mucinous cystic neoplasm** | NA | 4 (10.8) |
| | **Cyst** | NA | 2 (5.4) |
| | **Pancreatic neuroendocrine tumor (PNET)** | NA | 2 (5.4) |
| | **Pseudopapillary neoplasm** | NA | 1 (2.7) |
| | **Pancreatic intraepithelial neoplasia** | NA | 1 (2.7) |
| | **Microcystic serous cystadenoma** | NA | 1 (2.7) |
| **T Stage, n (%)** | **I** | 1 (1.7) | NA |
| | **II** | 2 (3.4) | NA |
| | **III** | 56 (94.9) | NA |
| | **IV** | 0 (0) | NA |
| **N Stage, n (%)** | **0** | 7 (11.9) | NA |
| | **I** | 51 (86.4) | NA |
| | **II** | 1 (1.7) | NA |
| **Histologic grade, n (%)** | **Well-differentiated** | 5 (8.5) | NA |
| | **Moderately differentiated** | 25 (42.4) | NA |
| | **Moderately-to-poorly differentiated** | 8 (13.5) | NA |
| | **Poorly differentiated** | 21 (35.6) | NA |
| **Treatment, n (%)** | **No neoadjuvant therapy** | 59 (100.0) | NA |

identify individual patients after data collection. All tissue samples were obtained from our tissue bank at the University of Florida Department of Surgery. Only patients with pathology confirmed PDAC naïve to neoadjuvant chemotherapy or radiation were included in the cancer group. Tissue was also collected separately from patients with pancreatitis, pancreatic cyst, and intraductal papillary mucinous neoplasm (IPMN) to serve as non-PDAC controls. All surgical specimens were collected by sharp dissection on a back table in the operating room. Pancreatic tissues were also collected from patients with chronic pancreatitis or other benign conditions that should not impact the resected pancreas, i.e., duodenal adenoma. Tissue used in the study was collected from distinct and separate patients with the diseases shown in Table 1. Samples were then transported on ice in Dulbecco's modified Eagle media supplemented with 10% fetal bovine serum (Lonza Group) and 1% antibiotic/antimycotic solution (Corning Inc.) to the lab immediately. Samples were flash frozen within 20 minutes of collection and stored at -80˚C for downstream soluble protein analysis. Separately, matching FFPE blocks were requested from our pathology core for TMA assembly.

## Tissue microarray assembly

Tissue was formalin-fixed, and paraffin embedded following resection. Representative formalin fixed paraffin-embedded blocks and tumor areas were selected by a pathologist (A.A) based

on H&E-stained slides. Briefly, a hollow needle was used to remove 2 mm diameter tissue cores from regions of interest. These tissue cores were then inserted into a recipient paraffin block forming the TMA. Assembled TMAs for CD3+ and CD8+ lymphocytes originate from the same tumors and tissue. TMAs were randomly assembled in duplicate and subsequently assessed by a pathologist for quality assurance before subjecting to cutting 4-micron slides for subsequent immunohistochemical analyses.

## Assessment of tissue microarray

Presence of CD3+ and CD8+ lymphocytes were evaluated by immunohistochemistry using anti-CD3 and anti-CD8 antibodies (Dako Omnis, Agilent). The project pathologist (A.A.) semi-quantitatively graded each core using traditional scoring methodology (scores ranging from 0–3). Scores between each TMA were calculated and compared to quantitative assessment. Slides of the stained TMAs were scanned using an Aperio Scanscope CS microscope and digital images were acquired for quantitative analysis. Quantification of positively stained cells was performed using the software package QuPath 0.3.2. [8,14] (see https://qupath.github.io/). Cores were only excluded if no analyzable tissue was present, or if PDAC was not confirmed for that core. Image type settings in QuPath were set to Brightfield (H-DAB). Total counted cells and positively stained cells (DAB) for each were tabulated using the positive cell count function in QuPath (analyze -> cell detection -> positive cell detection). Whole tissue cores were selected for analysis. Percentage of positive cells were calculated using positive cells / total cells *100 (%). TMAs were stained and assembled in duplicates, averages between two matching TMAs were used, when available. Quantification and grading were performed in a blinded fashion separate from survival analysis. A total of 99 individual cores in the PDAC group, and 36 non-PDAC cores in the CD3+ TMA were graded and quantitated across duplicates. In the CD8+ TMA, 90 individual PDAC cores and 37 non-PDAC cores were graded and quantitated across duplicates. The minor differences in sample sizes were due to quality-assurance (i.e. exclusion due to core integrity/damage across TMAs).

## Soluble protein analysis

At the time of processing, tissue was thawed and weighed. Tissue was sharply divided into small pieces and placed into 2-mL Lysing Matrix D tubes (MP Biomedicals, Santa Ana, California, USA). For every 30mg of tissue, 500 uL of cell lysis buffer (Cell Signaling Technology, Danvers, Massachusetts) with Protease/Phosphatase Inhibitor (Cell Signalling Technology) was added. Samples were then placed on ice for 2 minutes each cycle. Lysates were collected and centrifuges at 13,000 relative centrifugal force for 10 minutes. Supernatants were collected and analyzed for total protein concentration. Homogenates were probed for 41 unique analytes using a commercially available multiplex analysis per the manufactures protocol (catalog no. HCYTMAG-60K-PX41; Millipore Sigma) and as described previously [15]. The multiplex assay was selected for its large inflammatory panel and used as exploratory analysis as previously described [15]. Data were acquired with the MAGPIX System (Luminex Corp) and analyzed using MILLIPLEX Analyst 5.1 (Millipore Sigma). Protein concentrations were normalized to total protein concentrations to yield individual analyte concentrations. Of the 41 analytes, those that provided non-informative data (no detection) were excluded. Samples from the TMA and in the soluble protein analysis were matched and originate from the same patient. These samples were processed separately (i.e., formalin fixation/paraffin embedding for the TMA, and sharp frozen for soluble protein analysis) but represent paired tissue after pathological confirmation.

## Survival analysis

Overall survival for our patient cohort was retrospectively obtained. Date of death was obtained through retrospective chart verification to determine survival time from surgical resection. Kaplan-Meier tests were applied, and the patient cohort was evenly split between the lower 50th percentile, and upper 50th percentile for CD3+ and CD8+ cell count percentages. Overall survival was also evaluated with semi-quantitative grading, categorizing the groups based on their grade (0, 1, 2, 3). Mantel-Cox survival comparisons were also made between quantification and grading for CD3+ and CD8+ patients.

## Statistical analysis

All statistical analyses were performed in the software GraphPad Prism 9.4.1. Shapiro-Wilk tests of the lymphocyte count and inflammatory protein levels in each tumor was used to assess normality and distribution. Due to non-normally distributed data observed in both lymphocyte cell counts and soluble protein analysis, non-parametric tests were used. Mann-Whitney U test was used to compare cell counts and infiltrating lymphocyte levels across tissues, and Spearman correlation was used to assess associations. Multivariate regression was used to assess relationships between clinicopathologic characteristics (i.e., CD3+% and age) in addition to spearman correlations. Kaplan-Meier method was employed to evaluate differences in overall survival. Spearman correlation analyses were performed between the percent of CD3 +/CD8+ cells and all 41 analytes in paired tumors or tissue. Data is presented as medians. A two-tailed P-value <0.05 is considered statistically significant.

## Results

### Quantitative analysis of intratumoral CD3+ and CD8+ lymphocytes in PDAC

The constructed TMA (prior to quality assurance) consisted of 59 treatment naïve pancreatic cancers and 37 non-PDAC controls. Clinicopathological characteristics are noted in Table 1. TMAs were immunohistochemically stained for CD3+ lymphocytes and CD8+ lymphocytes in duplicates. Our pathologist reviewed each TMA for quality assurance and semi-quantitatively graded the staining intensity in a blinded fashion. For quantitative assessment, cell counting was performed using the software package QuPath [8]. Following quality assurance, 56 PDAC tumors and 21 non-PDAC samples were analyzed in the CD3+ TMA, and 59 PDAC tumors and 24 non-PDAC samples were analyzed in the CD8+ TMA. The analyzable non-PDAC group included pancreatitis (n = 12), IPMN (n = 3), and other tissue (3 mucinous cystic neoplasm, 2 pancreatic neuroendocrine tumor, 1 pseudopapillary neoplasm) in the CD3 TMA. And the analyzable non-PDAC group for the CD8 TMA included pancreatitis (n = 13), IPMN (n = 4), and other (4 mucinous cystic neoplasm, 1 pancreatic neuroendocrine tumor, 1 squamoid cyst, and 1 pseudopapillary neoplasm).

PDAC displayed noteworthy heterogeneity in its CD3+ and CD8+ populations. Serial sections of tumor resulted in varying levels of CD3+ (Fig 1a) and CD8+ (Fig 1b) cells. To account for this, we created duplicate TMAs for each core, and, when possible, averaged the percent positive between the groups. We noted that the distribution of CD3+ and CD8+ lymphocytes in the PDAC tumors were distributed relatively evenly (not clustered). Non-PDAC pancreas maintained high cell density characteristic of normal pancreas, as opposed to the less dense, desmoplastic reactions that forms in tumors (Fig 2a and 2c). We observed significant increases in CD3+ and CD8+ populations (by percentages, and absolute values) in PDAC tumors compared to non-PDAC control tissues with the exception of PDAC vs. IPMN, however, this may

**Original image**    **Positive cell quantification**

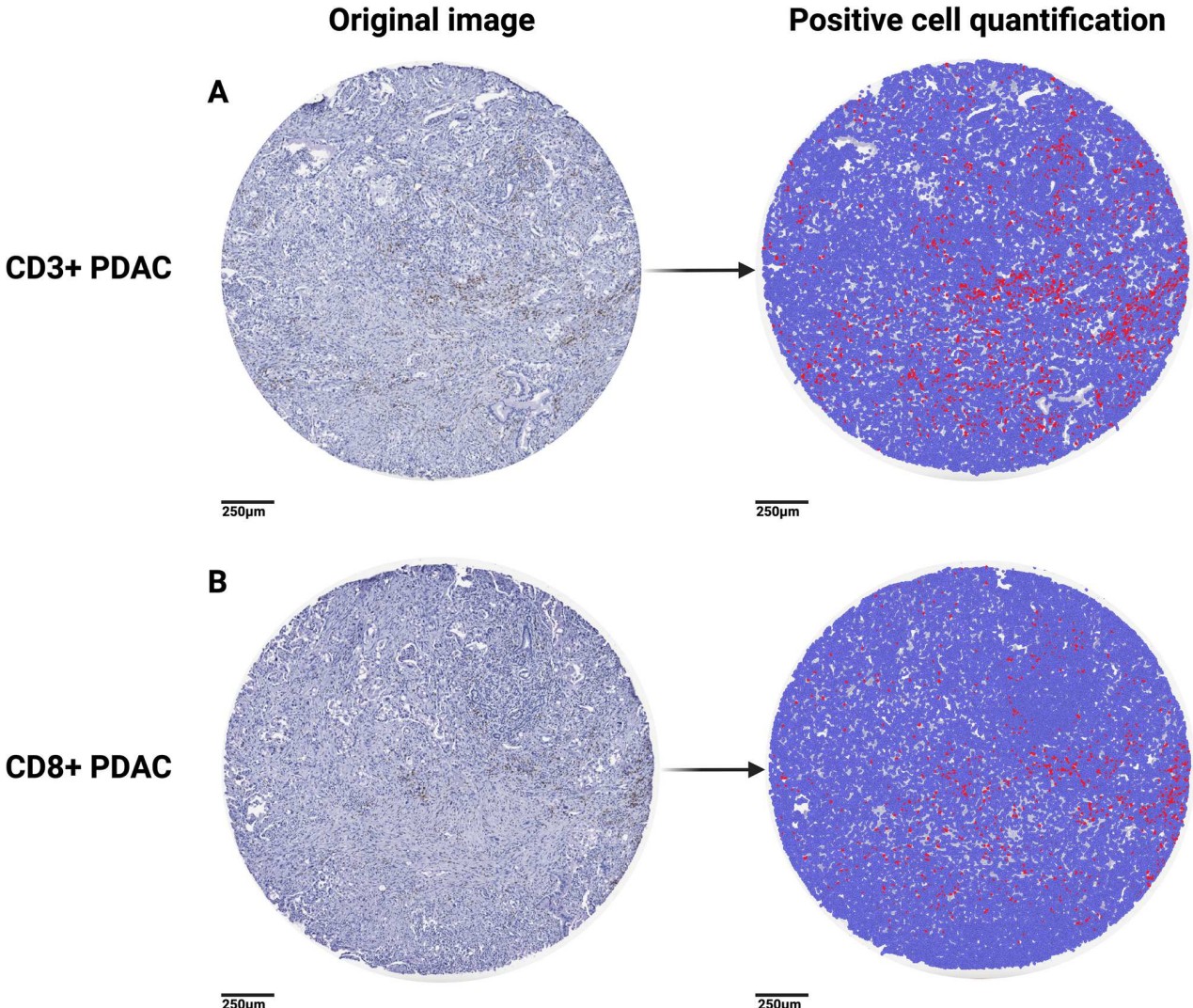

**Fig 1. Quantification of intratumoral CD3+ and CD8+ lymphocytes.** Representative samples of immunohistochemical staining of tumor cores, slices originate from the same tumor: (A) CD3+ PDAC and subsequent quantification, and (B) CD8+ PDAC and subsequent quantification. Blue-colored cells represent cells labelled negative, red-colored cells represent cells considered positive.

be due to limited sample size of the IPMN cohort (Fig 2b and 2d). As expected, Spearman r correlation of CD3+ and CD8+ revealed a significant correlation (Fig 2e). Analysis of covariates (CD3/CD8 infiltration, age, sex, histologic grade) in a multi-variable linear regression revealed no correlation, and poor model fit (S1 Table). This has also been shown in previous studies [16]. Spearman correlation of individual covariates (age, sex, histologic grade) also showed no correlation (S1 Table).

## Semi-quantitative and quantitative assessments and overall survival

We then compared quantitative analysis to semi-quantitative pathological grading to observe patterns between systematic scoring and quantitative techniques. As above, the TMAs were subject to evaluation by an independent pathologist and graded according to standard pathology guidelines (score 0–3). Individual cores were assessed, and duplicate TMA cores were

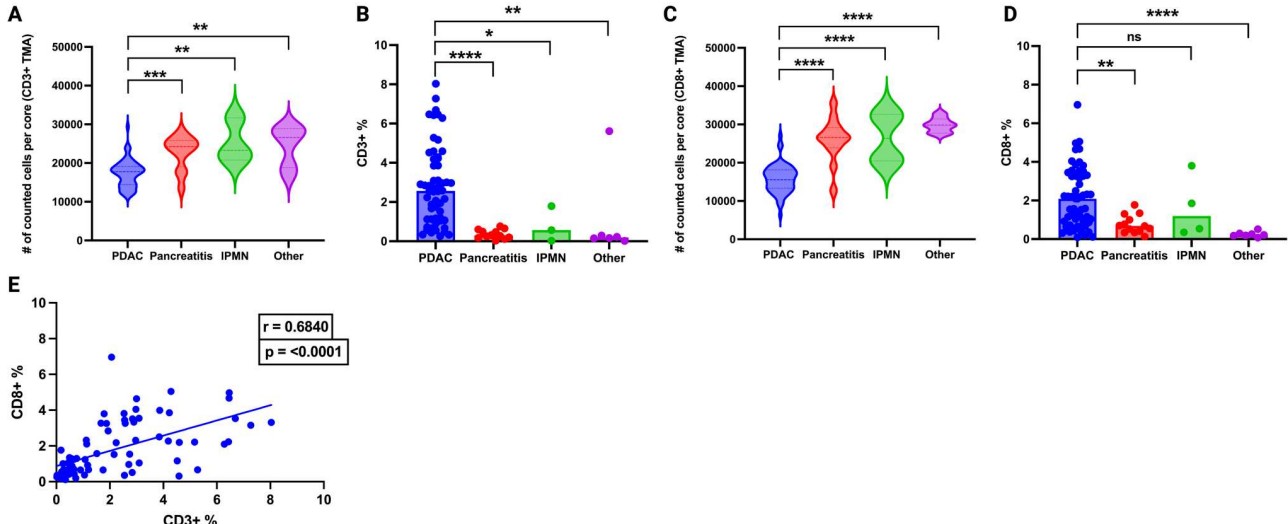

**Fig 2. Assessment of intratumoral CD3+ and CD8+ lymphocyte populations.** (A) Cell count of CD3+ TMA, PDAC (n = 56, median = 17785), Pancreatitis (n = 12, median = 24219), IPMN (n = 3, mean = 23315), Other (n = 6, mean = 26582), Mann Whitney U tests of PDAC vs Pancreatitis (p = 0.0003); PDAC vs IPMN (p = 0.0041); PDAC vs Other (p = 0.0031). (B) Prevalence of CD3+ lymphocytes in PDAC (n = 56, median = 2.565%), Pancreatitis (n = 12, median = 0.2950%), IPMN (n = 3, median = 0.5628%), Other (n = 6, median = 0.1919%). Mann-Whitney test of PDAC vs Pancreatitis (p = <0.0001), PDAC vs IPMN (p = 0.0434), PDAC vs other (p = 0.0031). (C) Cell count of CD8+ TMA, PDAC (n = 59, median = 15569), Pancreatitis (n = 13, median = 26580), IPMN (n = 4, median = 26355), Other (n = 7, median = 29877). (D) Prevalence of CD8+ lymphocytes in PDAC (n = 59, median = 2.090%), Pancreatitis (n = 13, median = 0.6800%), IPMN (n = 4, median = 1.195%), Other (n = 7, median = 0.2200%). Mann-Whitney test of PDAC vs Pancreatitis (p = 0.0018), PDAC vs IPMN (p = 0.5012), PDAC vs Other (p = <0.0001). (E) Correlation of CD3+ and CD8 + lymphocyte percentages of PDAC and non-PDAC in the TMAs, r = 0.684 and p = <0.0001.

used individually (as opposed to averaged values) to appropriately, and directly, assess grading and quantification conversion. Fig 3A and 3C visually demonstrate the impact of conversion from integer semi-quantification to continuous variable data from the digital quantification. To assess the prognostic value of CD3+ and CD8+ infiltration and compare differences in grading and quantification, we performed Kaplan-Meier tests. We split the cohort at the median, thus CD3+ or CD8+ low tumors from the bottom 50th percentile for cell count versus CD3+ or CD8+ high tumors from the top 50th percentile group. Survival by quantitative grading was split into three groups, 0, 1, 2. No tumors were graded as 3. In our survival comparison, grades of 0 were compared with the tumors with "low" infiltration, and grades of 1 or 2 were compared with tumors with "high" infiltration.

We did not observe a significant difference in overall survival between CD3+ low and high groups through quantitative technique, with a median survival of 273 days in CD3+ low tumors, and 642.5 days in CD3+ high tumors (p = 0.2184). Pathological grading for CD3 + cores demonstrated a statistical significance between 0 and 1 grades. CD3+ cores graded 0 demonstrated a median survival of 252 days, CD3+ cores graded 1 had a median survival of 778 days (p = 0.0017) (Fig 3b). No statistically significant differences in survival were found in grading vs. quantitative technique, CD3 low vs. 0 (p = 0.1907), and CD3 high vs. 1 (p = 0.7003) No cores in the CD3+ group were graded 2.

We observed a significant difference between CD8+ low and CD8+ high tumors, with a median survival of 240 days, and 1059 days, respectively (p = 0.0003). CD8+ tumors with grades 0, 1, 2, also demonstrated statistically significance, with a median overall survival of 240 days, 642 days, and 951 days, respectively (p = 0.0156) (Fig 3d). Comparison of grading and quantitative technique found no significant differences between CD8+ low vs. 0 (p = 0.7982), CD8+ high vs. 1 (p = 0.6887), and CD8+ high vs 2 (p = 0.5882). Taken together, this data

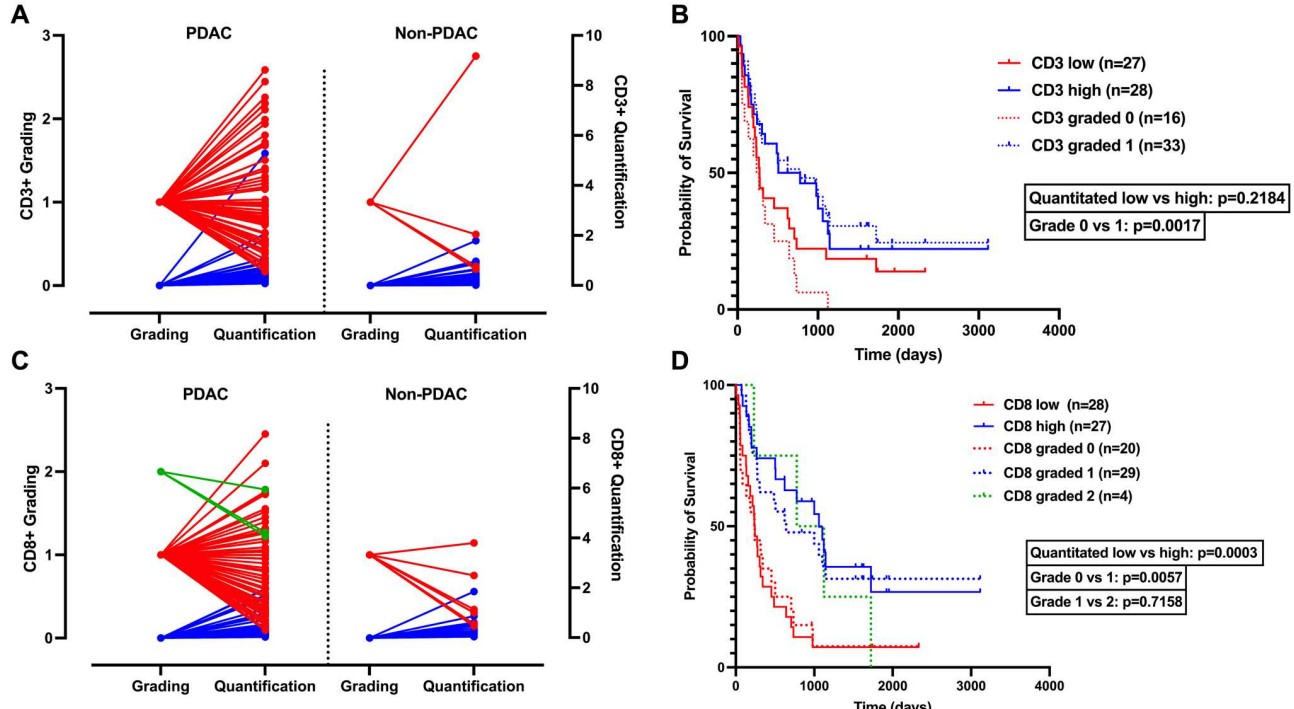

**Fig 3. Comparison of semi-quantitative and quantitative assessments of CD3+ and CD8+ lymphocytes in association with overall survival.** (A) Comparison of semiquantitative grading of cores with quantitative assessment for CD3+ staining (blue lines represent a grade of 0, red lines represent a grade of 1, green lines represent a grade of 2). In the CD3+ TMA, 99 individual cores in the PDAC group, and 36 cores in the non-PDAC group were graded and quantitated. (B) CD3+ overall survival comparison of quantitative methods vs grading method (CD3+ low = solid red, CD3+ high = solid blue, 0 = dashed red line, 1 = dashed blue line). CD3+ low tumors had a median survival of 273 days, and a median survival of 642.5 days in CD3+ high tumors (p = 0.2184). CD3+ cores graded 0 demonstrated a median survival of 252 days, CD3+ cores graded 1 had a median survival of 778 days (p = 0.0017). In comparing grading vs. quantitative technique, no statistically significant difference was found; CD3 low vs. 0 (p = 0.1907), and CD3 high vs. 1 (p = 0.7003). (C) Comparison of semi-quantitative grading of cores converted to quantitative assessment for CD8+ staining. In the CD8 + TMA, 90 individual cores and 37 non-PDAC cores were graded and quantitated. (D) CD8+ overall survival comparison of quantitative methods vs grading method. CD8+ low tumors had a median survival of 240 days, CD8+ high tumors had a median survival of 1059 days (p = 0.0003). CD8 + tumors with grade 0 vs 1 had a median survival of 240 days and 642 days (p = 0.0057), and grade 1 vs 2 had a median survival of 642 and 951 days (p = 0.7158). Grading vs. quantitative technique showed no significant differences, CD8+ low vs 0 (p = 0.7982), CD8+ high vs 1 (p = 0.6887), CD8 + high vs 2 (p = 0.5882).

shows how a quantitative approach may provide more precise, continuous data that does not necessarily result in superior prognostication in this relatively small sample set.

## Identification of an inflammatory protein signature in PDAC with increased CD3+ and CD8+ infiltration

Inflammatory signaling mediates the activation and recruitment of cells to the tumor site [17,18]. We hypothesized that CD3+ high tumors and CD8+ high tumors had distinct tumor microenvironment inflammatory milieus. To test this, we performed a multiplex analysis of 41 inflammatory analytes with subsequent Spearman correlation analyses to find associations between these analytes and levels of CD3+ (Fig 4a) or CD8+ (Fig 5a) infiltration in PDAC. We performed a principal component analysis (PCA) of CD3+ PDAC, pancreatitis, IPMN and other miscellaneous tissue. Principal component analysis (PCA) used 31 analytes, as data from 10 analytes (G-CSF, IL-9, IL-1B, IL-2, IL-3, IL-4, IL-5, MIP-1A, RANTES, and TNFB) were non-informative (Figs 4d and 5c). PCA visually demonstrates the diversity of inflammatory signatures and notable heterogeneity in PDAC, with some PDAC showing overlap between

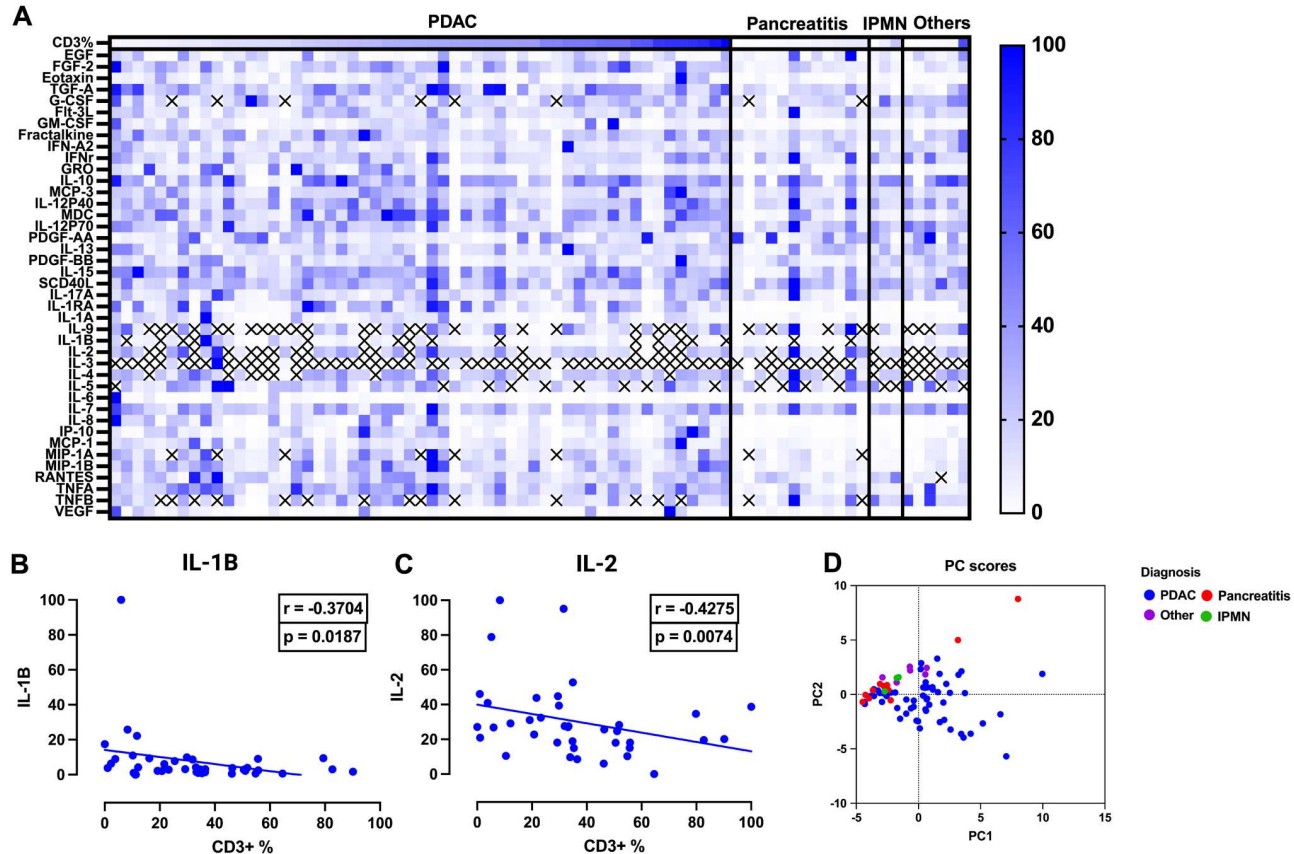

**Fig 4. Expression of cytokines by CD3+ infiltration in PDAC.** (A) Heatmap of 41 analytes in PDAC (n = 55) and non-PDAC tissue (Pancreatitis (n = 12), IPMN (n = 3), Other (n = 6)), expression levels were normalized to protein concentration (B, C): Spearman correlation analysis of IL-1B (r = -0.3704, p = 0.0187) and IL-2 (r = -0.4275, p = 0.0074) show decreased concentrations were significantly associated with increased CD3+ infiltration. (D) Principal component analysis plot of inflammatory signatures in different tissues, PCA was performed using 31 analytes, as 10 analytes (G-CSF, IL-9,IL-1B,IL-2,IL-3,IL-4,IL-5,MIP-1A,RANTES and TNFB) had limited detection across our patient cohort. PDAC (blue) is noted to be heterogenous in its inflammatory profile compared to non-PDAC tissue (Pancreatitis = red, Other = purple, IPMN = green).

IPMN and non-PDAC signatures. Out of 41 inflammatory proteins analyzed (Table 2) We found higher levels of IL-1B and IL-2 to be significantly correlated with lower CD3+ infiltration, r = -0.3704, -0,4275, with p = 0.0187, 0.0074, respectively (Fig 4b and 4c).

We next aimed to identify inflammatory proteins associated with CD8+ infiltration. PCA was performed using 31 analytes as mentioned above. The PCA similarly shows high variability in the presence of inflammatory proteins in tumors, and some overlap between non-PDAC (Fig 5c). Of the 41 inflammatory proteins assessed (Table 3), only IL-1B demonstrated statistical significance and was associated with decreased CD8+ infiltration (r = -0.4299, p = 0.0045) (Fig 5b).

## Discussion

In this work, we digitally measured intratumoral CD3+ and CD8+ lymphocytes and confirmed its effectiveness in the quantitative assessment of T-cell infiltration, which is predictive of overall survival in PDAC. PDAC demonstrated increased CD3+ and CD8+ infiltration compared to non-PDAC controls, and substantial heterogeneity. We found a significant increase in overall survival with high levels of CD8+ infiltration, but not CD3+ infiltration. We

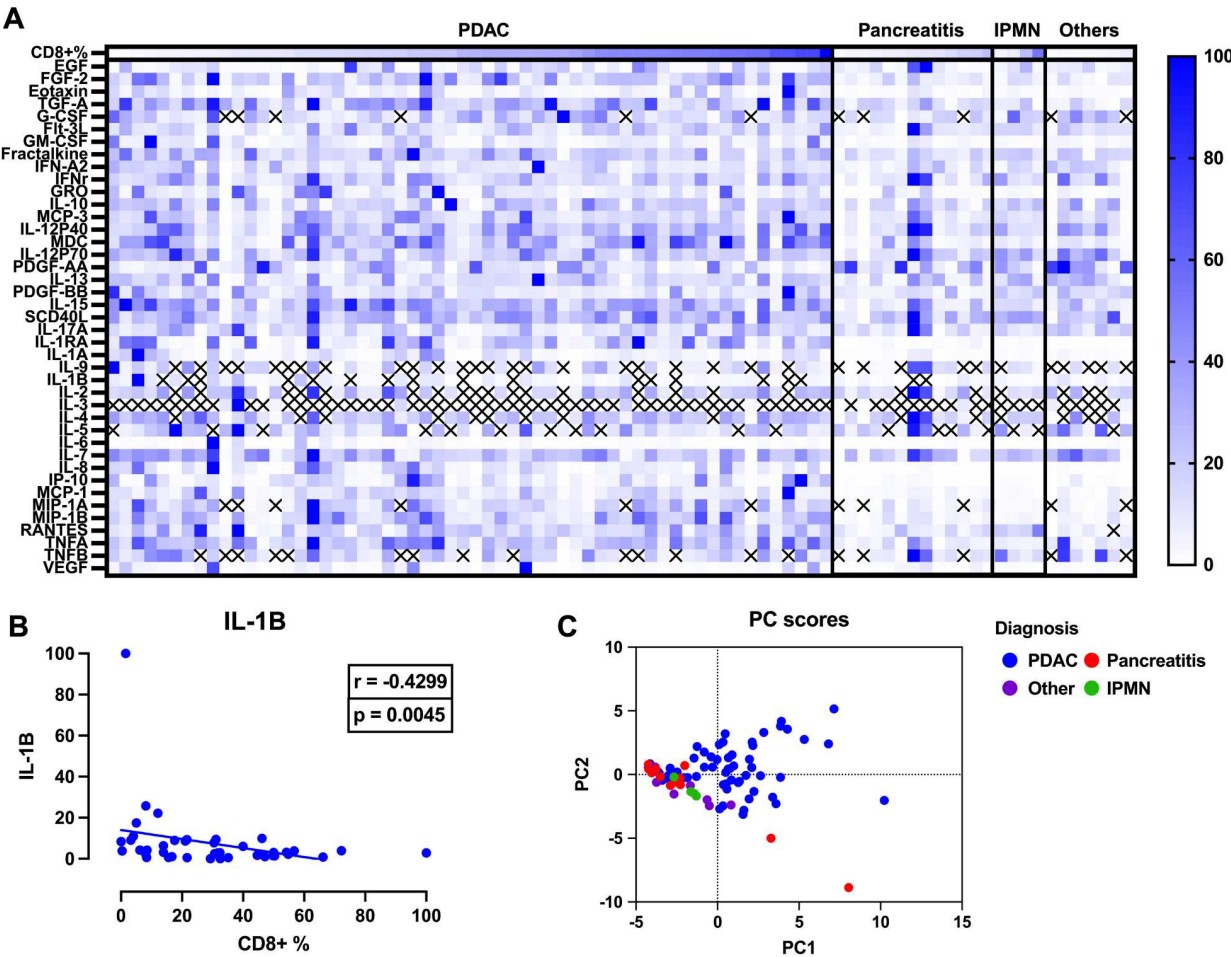

**Fig 5. Expression of cytokines by CD8+ infiltration in PDAC.** (A) Expression of 41 analytes in PDAC (n = 58) and non-PDAC (Pancreatitis (n = 13), IPMN (n = 4), Other (n = 7)) expression is normalized to protein concentration (B) Spearman correlation analyses showing increased expression of IL-1B significantly associated with decreased CD8+ infiltration (r = -0.4299, p = 0.0045). (C) Principal component analysis plot of inflammatory signatures in different tissues, PCA was performed using 31 analytes, as 10 analytes (G-CSF, IL-9,IL-1B,IL-2,IL-3,IL-4,IL-5,MIP-1A, RANTES and TNFB) had limited detection across our patient cohort. PDAC (blue) is noted to be heterogenous in its inflammatory profile compared to non-PDAC tissue (Pancreatitis = red, Other = purple, IPMN = green).

further demonstrated correlations between inflammatory protein signatures and levels of CD3 + and CD8+ infiltration.

Previous studies have used a range of approaches to evaluate the presence of CD3+ and CD8 + lymphocytes in tumors, including both quantitative and semiquantitative methods [19]. In this study, we contributed a standardized and reproducible method for quantifying CD3+ and CD8+ T-cells within a treatment-naïve patient cohort with PDAC. Semi-quantitative grading and quantitative approach through QuPath demonstrated similar outcomes in both prevalence of lymphocyte infiltration, and prognosis. The limited sample size in tumor or tissue graded 2 or 3 may limit the generalizability of these findings, and this analysis should be reproduced in tumors with expectantly higher lymphocyte infiltration, such as melanoma or microsatellite instable solid tumors. Through the identification of patients with objectively higher levels of CD8+ infiltration, we may be able to more effectively select patients who are most likely to benefit from immune-based therapies, such as checkpoint inhibitors or adoptive cell transfer.

**Table 2. Correlation of CD3+ infiltration and chemokines/cytokines.**

| Immune cell type | Cytokine/chemokine | # of XY pairs | Spearman r | 95% confidence | p-value |
|---|---|---|---|---|---|
| CD3 | EGF | 55 | -0.0008658 | -0.2736 to 0.2719 | 0.995 |
| CD3 | FGF-2 | 55 | 0.08615 | -0.1911 to 0.3507 | 0.5317 |
| CD3 | Eotaxin (CCL11) | 55 | -0.1139 | -0.3750 to 0.1640 | 0.4079 |
| CD3 | TGF-A | 55 | -0.01465 | -0.2863 to 0.2591 | 0.9155 |
| CD3 | G-CSF | 49 | 0.04684 | -0.2455 to 0.3314 | 0.7493 |
| CD3 | Flt-3L | 55 | 0.1027 | -0.1750 to 0.3652 | 0.4557 |
| CD3 | GM-CSF | 55 | -0.1841 | -0.4350 to 0.09331 | 0.1784 |
| CD3 | Fractalkine (CX3CL1) | 55 | -0.04192 | -0.3111 to 0.2335 | 0.7612 |
| CD3 | IFN-A2 | 55 | 0.01053 | -0.2630 to 0.2825 | 0.9392 |
| CD3 | IFNr | 55 | 0.08268 | -0.1945 to 0.3476 | 0.5484 |
| CD3 | GRO (CXCL1) | 55 | -0.1622 | -0.4165 to 0.1157 | 0.2368 |
| CD3 | IL-10 | 55 | 0.1274 | -0.1506 to 0.3867 | 0.3539 |
| CD3 | MCP-3 (CCL7) | 55 | -0.07446 | -0.3403 to 0.2024 | 0.589 |
| CD3 | IL-12P40 | 55 | 0.06674 | -0.2098 to 0.3334 | 0.6283 |
| CD3 | MDC (CCL22) | 55 | 0.08312 | -0.1940 to 0.3480 | 0.5463 |
| CD3 | IL-12P70 | 55 | -0.05887 | -0.3264 to 0.2174 | 0.6694 |
| CD3 | PDGF-AA | 55 | -0.01847 | -0.2898 to 0.2556 | 0.8935 |
| CD3 | IL-13 | 55 | 0.07727 | -0.1997 to 0.3428 | 0.575 |
| CD3 | PDGF-BB | 55 | -0.1454 | -0.4022 to 0.1326 | 0.2896 |
| CD3 | IL-15 | 55 | -0.1506 | -0.4067 to 0.1273 | 0.2723 |
| CD3 | SCD40L | 55 | 0.1591 | -0.1188 to 0.4139 | 0.246 |
| CD3 | IL-17A | 55 | -0.02734 | -0.2979 to 0.2472 | 0.8429 |
| CD3 | IL-1RA | 55 | 0.01053 | -0.2630 to 0.2825 | 0.9392 |
| CD3 | IL-1A | 55 | -0.1569 | -0.4120 to 0.1210 | 0.2526 |
| CD3 | IL-9 | 32 | -0.1276 | -0.4645 to 0.2416 | 0.4866 |
| CD3 | IL-1B | 40 | -0.3704 | -0.6173 to -0.05703 | 0.0187* |
| CD3 | IL-2 | 38 | -0.4275 | -0.6629 to -0.1152 | 0.0074* |
| CD3 | IL-3 | 6 | -0.6 | NA | 0.2417 |
| CD3 | IL-4 | 46 | -0.09504 | -0.3826 to 0.2093 | 0.5298 |
| CD3 | IL-5 | 46 | -0.1023 | -0.3888 to 0.2022 | 0.4987 |
| CD3 | IL-6 | 55 | -0.2247 | -0.4688 to 0.05114 | 0.099 |
| CD3 | IL-7 | 55 | 0.02027 | -0.2539 to 0.2914 | 0.8832 |
| CD3 | IL-8 (CXCL8) | 55 | -0.1544 | -0.4099 to 0.1236 | 0.2604 |
| CD3 | IP-10 (CXCL10) | 55 | 0.08947 | -0.1879 to 0.3536 | 0.516 |
| CD3 | MCP-1 (CCL2) | 55 | -0.1016 | -0.3642 to 0.1760 | 0.4605 |
| CD3 | MIP-1A (CCL3) | 49 | -0.02959 | -0.3159 to 0.2617 | 0.84 |
| CD3 | MIP-1B (CCL4) | 55 | 0.009596 | -0.2638 to 0.2816 | 0.9446 |
| CD3 | RANTES (CCL5) | 55 | -0.03896 | -0.3084 to 0.2363 | 0.7776 |
| CD3 | TNFA | 55 | -0.0386 | -0.3081 to 0.2366 | 0.7796 |
| CD3 | TNFB | 41 | -0.0899 | -0.3948 to 0.2329 | 0.5762 |
| CD3 | VEGF | 55 | -0.1509 | -0.4069 to 0.1270 | 0.2713 |

While PDAC has been shown to be resistant to current immunotherapeutics, recruitment and activation of T-cells into the tumor site may still be a viable approach [20,21]. Individual inflammatory molecules have been leveraged to guide immune cells into the tumor-microenvironment [22,23], however, a multifaceted approach to modifying inflammation, involving multiple inflammatory molecules, may result in improved immunogenicity in pancreatic

**Table 3. Correlation of CD8+ infiltration and chemokines/cytokines.**

| Immune cell type | Cytokine/chemokine | # of XY pairs | Spearman r | 95% confidence | p-value |
|---|---|---|---|---|---|
| CD8 | EGF | 58 | 0.1222 | -0.1482 to 0.3755 | 0.361 |
| CD8 | FGF-2 | 58 | 0.1275 | -0.1429 to 0.3802 | 0.3403 |
| CD8 | Eotaxin (CCL11) | 58 | -0.03522 | -0.2980 to 0.2325 | 0.793 |
| CD8 | TGF-A | 58 | -0.08884 | -0.3462 to 0.1810 | 0.5072 |
| CD8 | G-CSF | 52 | -0.001964 | -0.2824 to 0.2787 | 0.989 |
| CD8 | Flt-3L | 58 | 0.1486 | -0.1218 to 0.3984 | 0.2657 |
| CD8 | GM-CSF | 58 | -0.1146 | -0.3690 to 0.1557 | 0.3915 |
| CD8 | Fractalkine (CX3CL1) | 58 | -0.08213 | -0.3403 to 0.1875 | 0.5399 |
| CD8 | IFN-A2 | 58 | -0.1331 | -0.3851 to 0.1373 | 0.3191 |
| CD8 | IFNr | 58 | 0.03857 | -0.2293 to 0.3011 | 0.7737 |
| CD8 | GRO (CXCL1) | 58 | -0.1366 | -0.3881 to 0.1338 | 0.3065 |
| CD8 | IL-10 | 58 | 0.01781 | -0.2489 to 0.2820 | 0.8944 |
| CD8 | MCP-3 (CCL7) | 58 | -0.02572 | -0.2893 to 0.2415 | 0.848 |
| CD8 | IL-12P40 | 58 | 0.182 | -0.08777 to 0.4270 | 0.1714 |
| CD8 | MDC (CCL22) | 58 | 0.2436 | -0.02346 to 0.4783 | 0.0653 |
| CD8 | IL-12P70 | 58 | -0.1283 | -0.3809 to 0.1421 | 0.337 |
| CD8 | PDGF-AA | 58 | -0.1466 | -0.3967 to 0.1238 | 0.2722 |
| CD8 | IL-13 | 58 | 0.04411 | -0.2241 to 0.3061 | 0.7423 |
| CD8 | PDGF-BB | 58 | -0.199 | -0.4413 to 0.07026 | 0.1342 |
| CD8 | IL-15 | 58 | -0.1433 | -0.3939 to 0.1271 | 0.2832 |
| CD8 | SCD40L | 58 | 0.1694 | -0.1007 to 0.4162 | 0.2036 |
| CD8 | IL-17A | 58 | -0.0474 | -0.3091 to 0.2210 | 0.7238 |
| CD8 | IL-1RA | 58 | 0.1378 | -0.1326 to 0.3891 | 0.3023 |
| CD8 | IL-1A | 58 | -0.1898 | -0.4335 to 0.07977 | 0.1535 |
| CD8 | IL-9 | 35 | -0.244 | -0.5411 to 0.1073 | 0.1578 |
| CD8 | IL-1B | 42 | -0.4299 | -0.6544 to -0.1358 | 0.0045* |
| CD8 | IL-2 | 41 | -0.007753 | -0.3231 to 0.3091 | 0.9616 |
| CD8 | IL-3 | 6 | -0.3143 | NA | 0.5639 |
| CD8 | IL-4 | 49 | -0.08429 | -0.3645 to 0.2099 | 0.5647 |
| CD8 | IL-5 | 47 | -0.2104 | -0.4760 to 0.09037 | 0.1557 |
| CD8 | IL-6 | 58 | -0.1514 | -0.4008 to 0.1190 | 0.2567 |
| CD8 | IL-7 | 58 | -0.1971 | -0.4396 to 0.07227 | 0.1381 |
| CD8 | IL-8 (CXCL8) | 58 | -0.1671 | -0.4143 to 0.1030 | 0.2099 |
| CD8 | IP-10 (CXCL10) | 58 | 0.04427 | -0.2239 to 0.3062 | 0.7414 |
| CD8 | MCP-1 (CCL2) | 58 | -0.1098 | -0.3648 to 0.1604 | 0.4117 |
| CD8 | MIP-1A (CCL3) | 52 | 0.05182 | -0.2321 to 0.3276 | 0.7152 |
| CD8 | MIP-1B (CCL4) | 58 | 0.03916 | -0.2288 to 0.3016 | 0.7704 |
| CD8 | RANTES (CCL5) | 58 | 0.08416 | -0.1856 to 0.3421 | 0.5299 |
| CD8 | TNFA | 58 | 0.003045 | -0.2627 to 0.2684 | 0.9819 |
| CD8 | TNFB | 44 | -0.2149 | -0.4880 to 0.09650 | 0.1612 |
| CD8 | VEGF | 58 | -0.2038 | -0.4453 to 0.06531 | 0.1249 |

cancer [24]. In this study, we established connections between T-cells, effector T-cells and inflammatory molecules in order to identify potential candidates that could aid in the immune recognition of PDAC.

The recruitment and activation of immune cells into the tumor microenvironment is fundamental to the success of immunotherapy. Our findings, as well as others, illustrated the

importance of tumor-derived cytokines in lymphocyte recruitment [25–28]. Factors secreted by cancer cells, such as GM-CSF and G-CSF have been reported to be associated with the recruitment of leukocytes into the TME [26,29]. Additionally, cytokines have been identified as mediators of therapeutic resistance in PDAC [30]. These cytokines remodel ambient cell populations to create a tumor permissive environment. Infiltration of T-cells in solid tumors have been associated with superior survival, however, in PDAC, populations of intratumor T-cells are scarce and are often unable to mount an effective anti-tumor immune response [31]. IL-1B has been shown to promote tumor survival, proliferation, and metastatic potential [32]. IL-1B also influences the tumor microenvironment by enhancing desmoplasia and immune suppression in pancreas and breast cancer [32–34]. IL-2 is a cytokine responsible for the growth, proliferation, and survival of T-cells. IL-2 has been shown to play a vital role in the regulation of CD8+ T-cells to maintain their reactivity against tumor cells [35,36]. Our data suggests that IL-1B may play a more dominant role in regulating T-cell recruitment into the tumor microenvironment. We posit that IL-2's role is more nuanced, potentially contributing to the activation and exhaustion of T-cells or effecting other T-cell subtypes and immune cells [37]. A more precise method for remodeling the tumor microenvironment may be achievable though spatial analysis of the microenvironment with respect to the secretion of inflammatory proteins and lymphocyte involvement. We previously reported differences in the transcriptome of cancer-associated fibroblasts based on proximity to tumor [38], these changes may be mediated by surrounding inflammatory molecules. In immune-hot solid tumors that respond well to immune checkpoint inhibitors, such as melanoma, T-cell and other immune cells that are recruited into the tumor-microenvironment are essential to mounting an effective immune response [39]. We speculate that there are inflammatory characteristics of CD3+ or CD8 + abundant tumors that can be exploited with tumors with less immunogenicity.

The heterogeneity of the pancreatic tumor microenvironment has posed challenges for both treatment and study [40–42]. In our study, we aimed to minimize variability by only including treatment naïve PDAC patients. Yet, we still observed significant heterogeneity in both the CD3+ and CD8+ populations and in the levels of inflammatory molecules across samples. We reported limited detection of certain cytokines (G-CSF, IL-9, IL-1B, IL-2, IL-3, IL-4, IL-5, MIP-1A, RANTES, and TNFB), though these cytokines may still be useful for further investigation. While our exploratory analysis found significant associations between IL-1B and IL-2 and lymphocyte infiltration, there is a need for additional mechanistic study to exploit the inflammatory axis to induce immune response to tumor. Future studies with larger validation cohorts that are more representative of the patient population could greatly improve our understanding of inflammation in PDAC and other solid tumor malignancies. Assessing the location of lymphocytes and inflammatory proteins (i.e., central tumor or tumor margin) may also prove useful to increase infiltration and penetration of lymphocytes in solid tumors. The relatively small sample size, and the fact that the patient cohort used in our analysis consisted largely of T-stage III tumors, may limit generalizability of our findings. A larger sample size, from various stages of tumor progression that assess whole tumor may provide greater statistical power, reduce the risk of type II errors, and allow for additional subgroup analyses.

Cancer cells employ pro/anti-inflammatory cytokines and chemokines to suppress or promote an immune-rich microenvironment [43–45]. The presence or absence of these cytokines can dictate local cell populations and ultimately the host's response to malignancy. Here, we found that increased levels of IL-1B and IL-2 were associated with lowered CD3+ T-cell infiltration, and increased IL-1B was associated with decreased CD8+ T-cell populations. Solid-tumor malignancies, such as PDAC, have a particularly inflexible scaffolding in the form of stroma that add an additional physical barrier to limit immune-cell involvement and sensitivity to traditional therapeutics [46,47]. Ongoing clinical trials are utilizing cytokines as vehicles

to increase tumor immunogenicity and remodeling the tumor microenvironment [48]. Our study combines advances in intratumoral immune cell quantification and characterization of inflammatory components to advance our understandings of the PDAC tumor microenvironment.

## Supporting information

**S1 Table. Comparisons of infiltrating CD3+ and CD8+ lymphocytes and covariate analysis.**
(XLSX)

**S2 Table. Dataset of patient cohort, CD3+ and CD8+ lymphocyte quantification, and inflammatory assay.**
(XLSX)

## Acknowledgments

We would like to thank Dr. Dongtao Ann Fu, Cathy Yuping Sun, and the University of Florida Molecular Pathology Core (RRID:SCR_016601) for assistance with TMA assembly, staining, and scanning. We would also like to acknowledge the contributions of our patients and their families for enabling us to advance our understanding of pancreatic cancer.

## Author Contributions

**Conceptualization:** Gerik W. Tushoski-Alemán.

**Data curation:** Gerik W. Tushoski-Alemán, Kelly M. Herremans, Patrick W. Underwood, Ashwin Akki.

**Formal analysis:** Gerik W. Tushoski-Alemán, Ashwin Akki.

**Funding acquisition:** Jose G. Trevino, Steven J. Hughes.

**Investigation:** Gerik W. Tushoski-Alemán, Ashwin Akki, Song Han.

**Methodology:** Gerik W. Tushoski-Alemán, Ashwin Akki, Song Han, Steven J. Hughes.

**Project administration:** Jose G. Trevino, Song Han, Steven J. Hughes.

**Resources:** Kelly M. Herremans, Patrick W. Underwood, Jose G. Trevino, Steven J. Hughes.

**Supervision:** Song Han, Steven J. Hughes.

**Validation:** Gerik W. Tushoski-Alemán, Kelly M. Herremans, Song Han, Steven J. Hughes.

**Visualization:** Gerik W. Tushoski-Alemán, Kelly M. Herremans, Andrea N. Riner, Song Han, Steven J. Hughes.

**Writing – original draft:** Gerik W. Tushoski-Alemán, Kelly M. Herremans, Patrick W. Underwood, Ashwin Akki, Andrea N. Riner, Song Han, Steven J. Hughes.

**Writing – review & editing:** Gerik W. Tushoski-Alemán, Kelly M. Herremans, Patrick W. Underwood, Ashwin Akki, Andrea N. Riner, Song Han, Steven J. Hughes.

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
