## [Decision Letter · Decision Letter 0]

16 Oct 2023

PONE-D-23-25893Infiltration of CD3+ and CD8+ lymphocytes in association with inflammation and survival in pancreatic cancerPLOS ONE

Dear Dr. Hughes,

Thank you for submitting your manuscript to PLOS ONE. After careful consideration, we feel that it has merit but does not fully meet PLOS ONE’s publication criteria as it currently stands. Therefore, we invite you to submit a revised version of the manuscript that addresses the points raised during the review process.

We look forward to receiving your revised manuscript.

Kind regards,

Kenji Fujiwara, PhD, MD

Academic Editor

PLOS ONE

Journal Requirements:

**Additional Editor Comments:**

Dear Dr. Hughes.

The article was reviewed by three reviewers and all recommended revision and I agree with them.

Best regards,

Kenji Fujiwara

Academic editor

Reviewers' comments:

Reviewer's Responses to Questions

**Comments to the Author**

1. Is the manuscript technically sound, and do the data support the conclusions?

Reviewer #1: Yes

Reviewer #2: Partly

Reviewer #3: Yes

2. Has the statistical analysis been performed appropriately and rigorously? 

Reviewer #1: Yes

Reviewer #2: No

Reviewer #3: Yes

3. Have the authors made all data underlying the findings in their manuscript fully available?

Reviewer #1: Yes

Reviewer #2: Yes

Reviewer #3: Yes

4. Is the manuscript presented in an intelligible fashion and written in standard English?

Reviewer #1: Yes

Reviewer #2: Yes

Reviewer #3: Yes

5. Review Comments to the Author

Reviewer #1: Comments to the authors

The authors show that prognosis is inversely related to the amount of CD3 and CD8-positive lymphocytes infiltrating into pancreatic cancer tumors by examining pancreatic cancer tissues. These results are very interesting and provide useful information for future treatment development. Below are comments and questions for this paper.

Major Comments:

１．Which part of the tumor did CD8 and CD3 cells analyze? Central? Invasive area? How does it differ from lymph node metastases? Was there a difference in normal areas of the tumor margin?

２．In relation to 1., where are CD3 and CD8 located in the tumor? Where in the tumor is there a relationship to the location of the tumor, such as where more is associated with a better prognosis?

３．Doesn't CD3 and CD8-positive cell infiltration correlate with the tumors and patient conditions shown in Table 1? Is there no correlation between stage and tumor differentiation? I think a more detailed analysis is needed.

３．How were frozen and formalin samples used differently? Is it correct to understand that protein was extracted from frozen samples for analysis of inflammatory cytokines, and that formalin samples were prepared for analysis of CD3 and CD8 expression by creating an Array? Describe Method in a little more detail to make it easier for the reader.

４．"Benign" contains a variety of miscellaneous tissues. How about comparing just IPMN and Pancreatitis, which have a lot of numbers? I think there is also quite a difference between Pancreatitis and IPMN. I don't think IPMN can be called "Benign." Does the tissue of Pancreatitis mean a specimen removed as a PDAC but without cancer? Or do you mean non-cancer tissues in cancer cases? Please provide a detailed description of “Benign“ definition and the source of the tissue specimen.

5. Why does the Digital Quantitative have only two groups while the Semiquantitative has three? I think the results would be more convincing if both were divided into three groups.

６．If CD3 and CD8 are significantly correlated within the same samples, is it necessary to separate the inflammatory protein signature into CD3 and CD8? There's a strong possibility of a similar outcome. If there is a difference between CD3 and CD8, why the difference?

Minor comments:

１．I think TMA means Tissue microarray assembly, but there is no description.

Reviewer #2: In this manuscript, Tushoski-Aleman et al. provide an analysis of CD3+ and CD8+ T cell infiltration of treatment-naïve pancreatic ductal adenocarcinoma specimens, along with complementary analysis of 41 different soluble proteins in the tumor microenvironment. The experimental approach is solid overall, although additional methodological details are needed. The inclusion of benign pancreatic tissue from patients with pancreatitis, cysts, or papillary mucinous neoplasm) is a strength of the study design. This manuscript should be of interest to tumor immunologists and oncologists who treat pancreatic cancer patients.

Major concerns:

1. Statistical analyses should consider covariates (sex, age, histologic grade).

2. It is unclear why the quality assurance review resulted in differing numbers of samples for staining CD3 versus CD8 in the TMAs (lines 154-156). It would be better to report data from the samples that were used for both types of staining as well as soluble protein analyses, so that the statistical comparisons made regarding protein markers (ex: linear regression data shown in Figures 2e and 5) are from the same pool of tissue specimens. Currently, it is impossible to determine whether the same samples were used to evaluate CD8+ T cell staining and CD3+ T cell staining (Figure 2e) or CD8+ staining and the soluble proteins listed in Figure 5. Making this change would also permit one set of patient demographic data to be shown in Table 1 that represents the tissue donors used for all analyses. At present, the characteristics of the patient donors used for the CD8 vs CD3 TMA vs the soluble protein analyses are unclear. Although this change would reduce the sample size slightly, it would improve the quality of the manuscript and would enhance interpretation of results.

3. In Figure 3, the numbers of samples used for pathological grading versus QuPath quantitation do not match (ex: n = 49 for CD3 graded versus n = 55 for CD3 QuPath quantitation) for either CD3 or CD8 analyses. It appears that the point of this figure is to provide comparisons between the two evaluation techniques; thus it is critical that the same samples be used for evaluation using both methods. The data presented should be revised accordingly.

4. In Figure 3, it appears that the curves for pathological grading versus the quantitative QuPath technique (ex: grading scale 1 versus CD3 high QuPath score) yielded similar results (despite the issues noted in point #3). This is a potentially important finding, especially for researchers who do not have access to QuPath. These statistical comparisons should be performed and results commented upon in the Results and Discussion.

5. In Figures 4 and 5, associations between CD3 and soluble protein analytes, or CD8 and soluble protein analytes, should be performed on matched samples. Was this done or were different sets of samples used for each type of analysis? The Methods should be revised to clarify.

6. The text describing the PCA in Figures 4 and 5 must be amended to include the authors’ interpretation of the results obtained. It is not sufficient to say that a PCA was done.

7. The Discussion needs to summarize what is known about GRO, IL-1b, and MDC regarding their roles in tumor immunity and CD8+ T cell function, particularly in PDAC.

8. The Discussion needs to address how the levels of infiltration by CD3+ and/or CD8+ cells seen here compare with frequencies of these cells seen in other tumor types, such as melanoma or RCC that are considered to be immunologically “hot” and provide references. Doing so is needed to permit readers to contextualize the findings presented in this manuscript. Also, are the authors refuting prior studies characterizing PANC as immunologically cold? Why or why not?

Minor concerns:

1. In the Introduction, the authors state: “The design of this study improves upon previous studies that have relied on semi-quantitative assessment and included neoadjuvant treated PDAC.” References must be provided here.

2. The Methods are confusing as written. Line 96 states that samples were flash frozen, but the TMA assembly section states that FFPE samples were used (Line 99). The two collection methods are incompatible. Were portions of tissue flash frozen for soluble protein analysis whereas other portions of the same tissue were used for formalin fixation? Please clarify.

3. The description of Figure 2 results in the text should be modified to clarify that although the percentages of CD3+ and CD8+ cell populations were increased in PDAC vs benign, the total numbers of CD3+ and CD8+ cells were greater in benign tissues.

4. The insertion of Figure Legends at random points within the text of the Results section makes reading the manuscript challenging. Please move Figure Legends to the end of the manuscript, as is typically done.

5. The color scheme for the Kaplan Meier curves in Figure 3 makes it difficult to discern group categories, especially in panel D. The color scheme and image resolution should be improved to enhance readability.

6. The authors should show p values on the graphs (not just in the text) for the Kaplan Meier curves shown in Figures 3B and 3D.

7. In Figures 4 and 5, the authors should explain why QuPath low versus high CD3 categorization (instead of pathological grading) was used to perform linear regression against soluble protein analytes. This is puzzling since the pathological grading resulted in significant survival differences for both CD3 and CD8, whereas QuPath-derived categorization did not.

8. For soluble protein chemokines, the authors should use approved CCL or CXCL designations throughout the text (ex: CXCL1 instead of GRO).

9. Throughout, the figure resolution is poor, which makes reading legends, p values, and data difficult, even after zooming in.

10. The manuscript should be read carefully to correct multiple minor grammatical errors.

Reviewer #3: In the statistical Method, the author should first justify the distribution pattern of the data like any normality distribution test- Kolmogorov-Smirnov test / Shapiro-Wilk test / Anderson Darling test etc.

6. PLOS authors have the option to publish the peer review history of their article (what does this mean?). If published, this will include your full peer review and any attached files.

Reviewer #1: No

Reviewer #2: No

Reviewer #3: No

---

## [Author Response · Author response to Decision Letter 0]

8 Dec 2023

11.29.2023

Dr. Kenji Fujiwara,

Re: PONE-D-23-25893

Title: Infiltration of CD3+ and CD8+ lymphocytes in association with inflammation and survival in pancreatic cancer

We appreciate the constructive feedback provided by the reviewers regarding our manuscript. Their comments and suggestions provide additional clarity and enhance the quality of our work. Here, we present our response to the points raised by each reviewer. 

Reviewer 1 (Major comments): 

1. Which part of the tumor did we analyze? Central? Invasive area? How does it differ from lymph node metastases? Was there a difference in normal areas of the tumor margin? 

The pancreatic ductal adenocarcinomas (PDAC) tumors for the tissue microarray (TMA) were taken from the center of each tumor. 7/59 tumors had no lymph node involvement, 52/59 had lymph node involvement, this is shown in table 1 in the “N Stage” category. We did not evaluate normal areas of the tumor margin, as the central core of each tumor was used. We have added this to the limitations of our discussion section. 

2. In relation to 1, where are the CD3 and CD8 located in the tumor? Where in the tumor is there a relationship to the location of the tumor, such as where more is associated with a better prognosis 

In most tumors with significant infiltration, we observed that the the CD3+ and CD8+ were distributed relatively evenly across the cores. This may be because the center of the tumor was used for the TMA. Several studies have assessed spatial differences in T-cell activation and T-cell exhaustion in PDAC and found that T-cell proximity to tumor was associated with better overall survival. We discuss a spatial basis for understanding T-cell activation in our discussion. 

3. Doesn't CD3 and CD8-positive cell infiltration correlate with the tumors and patient conditions shown in Table 1? Is there no correlation between stage and tumor differentiation? I think a more detailed analysis is needed.

The vast majority of our patient cohort with PDAC were T-Stage III (94.9%). We conducted an additional multivariable analysis to assess associations between CD3 and CD8 infiltration, age, sex, and histologic grade. This is now shown in figure 2F and 2G, which show no significant correlation to the model, and the lack of differently staged PDAC is noted in our discussion.

4. How were frozen and formalin samples used differently? Is it correct to understand that protein was extracted from frozen samples for analysis of inflammatory cytokines, and that formalin samples were prepared for analysis of CD3 and CD8 expression by creating an Array? Describe Method in a little more detail to make it easier for the reader.

For the cytokine/chemokine assay, tissue was flash-frozen within 20 minutes after resection. The rest of the tissue of interest was taken to the pathology core for processing in paraffin and diagnostic confirmation. This tumor was used as a donor block for the TMA, and the area of interest was confirmed by a pathologist. This has been clarified in the methods section – that the flash frozen tissue was used for the cytokine/chemokine assay, and the FFPE tissue was used to assemble the TMA and assess CD3+ and CD8+ lymphocyte infiltration, both from the same patient.

5. “Benign” contains a variety of miscellaneous tissues. How about comparing just IPMN and Pancreatitis, which have a lot of numbers? I think there is also quite a difference between pancreatitis and IPMN. I don’t think IPMN can be called “benign.” Does the tissue of pancreatitis mean a specimen removed as PDAC but without cancer, or do you mean non-cancer tissues in cancer cases? Please provide a detailed description of “Benign” definition and the source of the tissue specimen.

We agree that the “benign” label may not be representative of the patient cohort, especially in IPMNs. In figure 2, we have separated the previously designated “benign” tissue into pancreatitis, IPMN and other miscellaneous tissue. The “Non-PDAC” and other group is now described explicitly in the introduction of the results. We have also relabeled the PCA to distinguish the groups (PDAC, pancreatitis, IPMN, other), which now show the heterogeneity of PDAC. Figure 3A and 3C have also been amended to show PDAC and Non-PDAC. Tissue specimens were collected from separate patients, i.e., PDAC is only from PDAC tumor from one patient, and pancreatitis tissue (or other diagnoses) are each from distinct and separate patients. 

6. Why does the Digital Quantitative have only two groups while the semi-quantitative has three? It would be more convincing if both were divided into three groups. 

While we agree that it may be optimal to divide the quantification cohort in to three cohorts might represent the grading system better, only four samples were graded two (and none grade 2 in the CD3+), with the vast majority graded 0 or 1. The small sample size in the grade 2 arm would not allow for statistical comparison between quantification and grading and offer a visual representation as the they appear to be in the upper bound of the quantification set. We also now address the fact that there were limited cores graded 2 in our discussion. 

7. If CD3 and CD8 are significantly correlated within the same samples, is it necessary to separate the inflammatory protein signature into CD3 and CD8? There is a strong possibility of a similar outcomes. If there is a difference between CD3 and CD8, why the difference?

 We separated CD3 and CD8 to identify any difference in the inflammatory signature of PDAC with high CD3 or high CD8. After refining our statistical analyses (as suggested by reviewer 3), we found that higher levels of IL-1B and IL-2 were associated with decreased CD3+ infiltration, but only IL-1B to be associated with decreased CD8+ infiltration. This may suggest that CD8+ t-cells can be specifically drawn into tumors by removing or introducing these molecules into the tumor microenvironment. We agree that this may not be clear from the manuscript and have added this, along with contextual function of these molecules in the discussion section.

Reviewer 1 (Minor comments): 

1. I think TMA means Tissue microarray assembly, but there is no description.

In the abbreviations section, we have added TMA to mean tissue microarray. 

Reviewer 2 (Major comments): 

1. Statistical analyses should consider covariates (sex, age, histologic grade)

We have added a multivariate analysis of covariates included sex, age, histologic grade (Figure 2f, 2g) we found that there were no correlations with CD3 or CD8 infiltration, and sex/age/histologic grading in a multivariate model. Other groups have similarly found no significant correlation between covariates and CD3/CD8 infiltration.

2. It is unclear why the quality-assurance review results in differing numbers of samples for staining CD3 versus CD8 in the TMAs. It would be better to report data from the samples that were used for both types of staining as well as soluble protein analyses, so that the statistical comparisons made regarding protein markers (ex. Linear regression data shown in figures 2e and 5) are from the same pool of tissue specimens. Currently, it is impossible to determine whether the same samples were used to evaluate CD8+ T cell staining and CD3+ T cell staining (Figure 2w) or CD8+ staining and the soluble proteins listed in Figure 5. Making this change would also permit one set of patient demographic data to be shown in table 1 that represents the tissue donors for all analyses. At present, the characteristics of the patients’ donors used for the CD8 vs CD3 TMA vs the soluble protein analyses are unclear. Although this change would reduce the sample size slightly, it would improve the quality of the manuscript and enhance the interpretation of results.

We agree that it may not be clear why there are differences in sample sizes across analyses. There are two duplicate TMAs of the same tumors for CD3+ and CD8+ staining. This was done to capture potential heterogeneity of the tumor along the z-axis. The tumors and tissue that were quantitatively analyzed for CD3+ and CD8+ cells originate from the same tumors/tissue used for the cytokine assay. This has now been clarified in the methods section. 

3. In figure 3, the number of samples used for pathological grading versus QuPath quantitation do not match (i.e. n=49 CD3 graded, n=55 for QuPath quantitation) for either CD3 and CD8 analyses. It appears that the point of this figure is to provide comparisons between the two evaluation techniques; thus it is critical that the same samples be used for evaluation using both methods. The data presented should be revised accordingly.

We apologize and understand the confusion between sample sizes. The quantitative and grading comparison is done by using the individual cores on the duplicate TMAs. We only included samples with a matched quantification and grading. In figure 2, the percent positive was averaged across duplicate TMAs. In figure 3 we used the individual cores and their respective gradings to directly compare quantification and grading. In the methodology and figure legend of the manuscript, the exact number of individual cores assessed is now shown for additional clarity, along with a clarifying statement describing why there are differences in sample sizes in different TMAs. 

4. In figure 3, it appears that the curves for pathological grading versus the quantitative QuPath technique (ex: grading score 1 versus CD3 high QuPath score) yielded similar results (despite the issues noted in point 3). This is a potentially important finding especially for researchers who do not have access to QuPath. These statistical comparisons should be performed, and results commented upon in the results and discussion. 

We have now directly compared the individual arms in the quantitative and grading survival curves; CD3 low vs grade 0, CD3 high vs grade 1; CD8 low vs grade 0, CD8 high vs grade 1, CD8 high vs grade 2. Mantel-cox comparison showed no statistically significant differences between any of the individual groups. We have added this to the results and discussion section. 

5. In figures 4 and 5, associations between CD3 and soluble protein analytes, or CD8 and soluble protein analytes, should be performed on matched samples. Was this done or were different sets of samples used for each type of analysis. The methods should be revised to clarify. 

The same tumors/tissue used in the TMA were used in the soluble protein analysis. And only tissue paired with the assay were used in our analysis. We apologize for any confusion and have clarified this in methods section. 

6. The text describing the PCA in figures 4 and 5 must be amended to include the authors’ interpretation of the results obtained. It is not sufficient to say that a PCA was done. 

We have edited the TMA to show different groups of tissue, as suggested by reviewer 1. It now shows the diverse and varied inflammatory signature of PDAC in comparison to pancreatitis, IPMNs and miscellaneous other pancreatic tissue. 

7. The discussion needs to summarize what is known about GRO, IL1B and MDC regarding their roles in tumor immunity and CD8+ T cell function, particularly in PDAC

After review with our biostatistician and re-analyses of the data using non-parametric tests (see details in response to reviewer 3), we found that higher levels of IL-1B and IL-2 were associated with decreased CD3+ infiltration, but only IL-1B to be associated with decreased CD8+ infiltration. The manuscript now reflects these changes throughout, and we now discuss the contextual functions of IL-1B and IL-2 in our discussion.

8. The discussion needs to address how the levels of infiltration by CD3+ and/or CD8+ seen here compare with frequencies of these cells seen in other tumor types, such as melanoma or RCC that are considered to be immunologically “hot” and provide references. Doing so is needed to permit readers to contextualize the findings presented in the manuscript. Also are the authors refuting prior studies characterizing PANC as immunologically cold? Why or why not?

We agree that this would be a valuable addition to the discussion section to compare to immunologically hot vs. cold tumor types. We have added in the discussion melanoma, and tumors with microsatellite instability. 

Reviewer 2 (Minor comments): 

1. In the introduction, the authors state: “The design of this study improves upon previous studies that have relied on semi-quantitative assessment and included neoadjuvant treated PDAC.” References must be provided here. 

We have added references to studies that use semi-quantitative assessment and advantages/limitations of this method.

2. The methods are confusing as written. Line 96 states that samples were flash frozen, but the TMA assembly section states that FFPE were used (line 99). The two collection methods are incompatible. Were portions of tissue flash frozen for soluble protein analysis whereas other portions of the same tissue were used for formalin fixation? Please clarify. 

We have clarified in our methods that tissue was separately dissected for soluble protein analysis, and the TMA was assembled with FFPE methods with the guidance of a pathologist. 

3. The description of figure 2 results in the text should be modified to clarify that although the percentages of CD3+ and CD8+ cell populations were increased in PDAC vs. benign, the total numbers of CD3+ and CD8+ cells were greater in benign tissues.

The absolute values of CD3+ and CD8+ infiltration was higher compared to non-PDAC. We agree that this should be added to the results section.

4. The insertion of figure legends at random points within the text of the results section makes reading the manuscript challenging. Please move figure legends to the end of the manuscript as is typically done. 

We have moved the figure legends to the end of the manuscript.

5. The color scheme for the Kaplan Meier curves in figure 3 makes it difficult to discern group categories, especially in panel D. The color scheme and image resolution should be improved to enhance readability.

We have made the lines on the Kaplan Meier curves thicker and the color scheme more visible. All figures should now be at 1200dpi. 

6. The authors should show p-values on graphs (not just in text) for the Kaplan Meier curves shown in Figures 3B and 3D. 

We now show the p-value of Mantel-Cox test between curves in figure 3B and 3D. 

7. In figures 4 and 5, authors should explain why QuPath low versus high CD3 categorization (instead of pathological grading) was used to perform linear regression against soluble protein analytes. This is puzzling since the pathological grading results in significant survival difference for both CD3 and CD8 whereas QuPath-derived categorization did not.

We decided to use quantification as we have shown in figure 3 that the results are comparable and quantitative technique may pick up on more nuance between tumor samples. While pathological grading did show significant survival differences for both CD3 and CD8, the objective of employing QuPath-derived categorization was to explore potential subtleties in protein expression levels that may not necessarily translate immediately into observable survival differences. Additionally, the staining was very specific to CD3+ and CD8+ cell markers which may make it more appropriate to use quantitative technique. We also did not find grading to be significantly different between CD8 graded 1 vs. 2.

8. For soluble protein chemokines, the authors should use approved CCL or CXCL designation throughout the text (Ex. CXCL instead of GRO.)

The approved designations have been added to GRO (CXCL1), IP10 (CXCL10), and MDC (CCL22), and all other chemokines in table 2 and 3. 

9. Throughout, the figure resolution is poor, which makes reading legends, p-values, and data difficult, even after zooming in.

Figures are now uploaded at 1200dpi, and figures have been modified to ensure readability of statistical analyses.

10. The manuscript should be read carefully to correct multiple minor grammatical errors.

All authors have reviewed the manuscript and grammatical errors have been fixed for clarity. 

Reviewer 3 (Major and minor comments):

1. In the statistical method, the authors should first justify the distribution pattern of the data like any normality distribution test – Kolmogorov-Smirnov test/ Shapiro-Wilk test/ Anderson Darling test. Etc. 

To address normality of our data, we performed Shapiro-Wilk tests, the majority of data was considered non-normal despite log and square transformation attempts (as would be expected in a heterogenous cancer like pancreatic cancer). After reviewing with our biostatistician, we applied Mann-Whitney U Tests, and Spearman correlations for comparisons and associations respectively. This is now shown in our methods and the results reflected throughout the manuscript.

---

## [Decision Letter · Decision Letter 1]

4 Jan 2024

Infiltration of CD3+ and CD8+ lymphocytes in association with inflammation and survival in pancreatic cancer

PONE-D-23-25893R1

Dear Dr. Hughes,

We’re pleased to inform you that your manuscript has been judged scientifically suitable for publication and will be formally accepted for publication once it meets all outstanding technical requirements.

Kind regards,

Kenji Fujiwara, PhD, MD

Academic Editor

PLOS ONE

Additional Editor Comments (optional):

Dear Dr. Hughes.

Thank you for revising your manuscript appropriately. All reviewers and I agreed to the acceptance.

Yours sincerely,

Kenji Fujiwara

Academic editor

Reviewers' comments:

Reviewer's Responses to Questions

**Comments to the Author**

1. If the authors have adequately addressed your comments raised in a previous round of review and you feel that this manuscript is now acceptable for publication, you may indicate that here to bypass the “Comments to the Author” section, enter your conflict of interest statement in the “Confidential to Editor” section, and submit your "Accept" recommendation.

Reviewer #1: All comments have been addressed

Reviewer #2: All comments have been addressed

Reviewer #3: All comments have been addressed

2. Is the manuscript technically sound, and do the data support the conclusions?

Reviewer #1: Yes

Reviewer #2: Yes

Reviewer #3: Yes

3. Has the statistical analysis been performed appropriately and rigorously? 

Reviewer #1: Yes

Reviewer #2: Yes

Reviewer #3: Yes

4. Have the authors made all data underlying the findings in their manuscript fully available?

Reviewer #1: Yes

Reviewer #2: Yes

Reviewer #3: Yes

5. Is the manuscript presented in an intelligible fashion and written in standard English?

Reviewer #1: Yes

Reviewer #2: Yes

Reviewer #3: Yes

6. Review Comments to the Author

Reviewer #1: Authors politely respond to most comments and questions, and correctly changes the text. I think it became a useful and easy-to-understand paper for readers

Reviewer #2: The manuscript is much improved. All of my concerns have been sufficiently addressed during revision.

Reviewer #3: (No Response)

7. PLOS authors have the option to publish the peer review history of their article (what does this mean?). If published, this will include your full peer review and any attached files.

Reviewer #1: **Yes: **Toshio Fujisawa

Reviewer #2: No

Reviewer #3: No

---

## [Editor Report · Acceptance letter]

1 Feb 2024

PONE-D-23-25893R1 

PLOS ONE

Dear Dr. Hughes, 

I'm pleased to inform you that your manuscript has been deemed suitable for publication in PLOS ONE. Congratulations! Your manuscript is now being handed over to our production team.

Kind regards, 

on behalf of

Dr. Kenji Fujiwara 

Academic Editor

PLOS ONE